# “Fishing and Hunting”—Selective Immobilization of a Recombinant Phenylalanine Ammonia-Lyase from Fermentation Media

**DOI:** 10.3390/molecules24224146

**Published:** 2019-11-15

**Authors:** Evelin Sánta-Bell, Zsófia Molnár, Andrea Varga, Flóra Nagy, Gábor Hornyánszky, Csaba Paizs, Diána Balogh-Weiser, László Poppe

**Affiliations:** 1Department of Organic Chemistry and Technology, Budapest University of Technology and Economics, 1111 Budapest, Hungary; bell.evelin@mail.bme.hu (E.S.-B.); molnar.zsofia@mail.bme.hu (Z.M.); nflo222@gmail.com (F.N.); hornyanszky@mail.bme.hu (G.H.); 2Fermentia Microbiological Ltd., 1405 Budapest, Hungary; 3Institute of Enzymology, Research Center for Natural Sciences, Hungarian Academy of Science, 1117 Budapest, Hungary; 4Biocatalysis and Biotransformation Research Centre, Faculty of Chemistry and Chemical Engineering, Babeş-Bolyai University of Cluj-Napoca, 400028 Cluj-Napoca, Romania; avarga870126@gmail.com (A.V.); paizs@chem.ubbcluj.ro (C.P.); 5SynBiocat Ltd., 1172 Budapest, Hungary; 6Department of Physical Chemistry and Materials Science, Budapest University of Technology and Economics, 1111 Budapest, Hungary

**Keywords:** IMAC, selective enzyme immobilization, phenylalanine ammonia-lyase, magnetic nanoparticles

## Abstract

This article overviews the numerous immobilization methods available for various biocatalysts such as whole-cells, cell fragments, lysates or enzymes which do not require preliminary enzyme purification and introduces an advanced approach avoiding the costly and time consuming downstream processes required by immobilization of purified enzyme-based biocatalysts (such as enzyme purification by chromatographic methods and dialysis). Our approach is based on silica shell coated magnetic nanoparticles as solid carriers decorated with mixed functions having either coordinative binding ability (a metal ion complexed by a chelator anchored to the surface) or covalent bond-forming ability (an epoxide attached to the surface via a proper linker) enabling a single operation enrichment and immobilization of a recombinant phenylalanine ammonia-lyase from parsley fused to a polyhistidine affinity tag.

## 1. Introduction

### 1.1. Importance of Biocatalysts

The use of enzyme-based biocatalysts for biotransformations is getting more and more common both in research and in industrial processes [1,2]. This phenomenon is understandable considering the need of enantiomerically pure compounds in modern drug manufacturing and the pursuit of environmental protection. Biocatalysts allow to accomplish stereo-, regio- or enantioselective reactions under mild conditions in an easy and environmentally friendly way. Biocatalysts often provide reaction pathways at ambient temperature, without using strong bases, acids, expensive metal-containing catalyst or difficult to regenerate, harmful solvents. Beyond the advantageous properties of proteins as biocatalysts, knowledge on their structure and mechanism of action is also very important because several drugs and pesticides target various proteins in living organisms.

#### 1.1.1. Whole Cells and Isolated Enzymes as Biocatalysts

In the food industry, microorganisms have already been used without knowing their mechanism. Yeasts have been catalyzing the fermentation processes of wine, yoghurt or leavened bread production since thousands of years. Nowadays, we are using whole cells or their enzymes more consciously for many purposes in research and industry as well [2,3,4].

In the pharmaceutical and fine chemicals industries, the whole cells and enzymes have an important role as biocatalyst in selective biotransformations resulting in simpler building blocks [4,5,6,7,8,9] and more complex compounds like alkaloids [10] or steroids [11,12,13,14]. It is a common feature of the chiral active pharmaceutical ingredients that just one of the enantiomers has the desired effect. Due to their homochiral nature, enzyme-based biocatalysts are capable of providing the desired enantiomer through kinetic resolution, dynamic kinetic resolution or enantiotope selective biotransformations. The advantage of the last two methods is that the undesirable enantiomer of a racemate can be utilized via a racemization step or it is not even forming in the biotransformation. During the preceding decades, several valuable biotransformations have been implemented: resolution processes via hydroxyl or amine functional groups [15,16,17,18,19,20,21,22], oxidations [23,24,25] and reductions [26,27,28,29] in one step or in cascade reactions [30,31,32], production of amines with transaminases [30,32,33,34], imine reductases [28,35] or amine dehydrogenase [34,36] and amino acid formations [37,38,39]. Most of these reactions have industrial importance not only in pharmaceutical field [3,40] but in food [41,42,43,44,45] and fine chemical industries [46,47,48] or in modern analytics and diagnostic technologies as well [43,49,50,51,52,53]. Furthermore, biocatalysis has gained more and more importance in fuel industry regarding biodiesel production with lipases [54,55], in polymer industry to mediate biocatalyzed polymerizations [56,57,58] or in environmental processes to degrade various polymers or biopolymers by hydrolysis [59,60,61].

#### 1.1.2. Biocatalyst Production by Fermentation, Recombinant Techniques

The developments in molecular genetics, directed evolution, structure determination and bioinformatics have led to an enormous increase in knowledge, experience and new possibilities in fermentation techniques, protein structure manipulation and enzyme mechanism determination. Thanks to modern biotechnology, the production of a novel recombinant protein in an appropriate host system is a routine process in a well-equipped laboratory [62].

Nature provides a large number of enzymes to implement a great variety of reactions [63]. In addition to the natural properties and functions of wild type enzymes, their activity, substrate scope, and stability can be evolved further with protein engineering methods. Modification of the amino acid sequence randomly or consciously, based on the researcher experiences or bioinformatics-based information is a frequent method to improve the efficiency of biocatalysts. For the efficient production of proteins mostly bacterial or yeast recombinant hosts are used [63]. The DNA sequence coding the protein is delivered to the producing cells as part of a vector (e.g., plasmid, phage or artificial chromosome). Beyond the mandatory sections which are essential for the protein expression, vectors can contain other components like antibiotic resistance gene sequences and tag-coding sections. Antibiotic resistance allows the selective survival of the vector containing cells, whereas different tags are useful in the purification of the protein [64,65,66]. Recombinant protein expression should be induced, in this way higher protein amounts can be obtained even if the resulting protein is toxic for the cell. In case of bacterial expressions, usually the *lac* operon and isopropyl β-d-1-thiogalactopyranoside (IPTG, being a molecular mimic of the lactose metabolite, allolactose) as inducer of the gene regulation system are used [66]. During the upstream process even hundred-liter fermenter size can be achieved step by step starting from a small colony.

#### 1.1.3. Downstream Processes for Enzyme Production

After the cell culture has reached the desired state, the cells are harvested by centrifugation. The easier case of enzyme production is when the target enzyme is secreted, thus it can be separated from the sedimented cells [67]. Even in this case, a number of purification steps are needed to obtain pure enzyme. It is more complicated in the case of intracellular enzymes when after the cell disruption the enzyme has to be isolated from the cell debris and many other proteins.

Usually, the protein products are expressed intracellularly in the host cells, where they can be occasionally accumulated in inclusion bodies. In order to gain the desired protein, the host cells have to be disrupted by using mechanical methods like high pressure homogenization, bead mills or ultrasonication, or other physical, enzymatic or chemical methods disintegrating the cell wall [68,69,70]. After cell disruption, the protein containing supernatant can be separated from the cell debris by centrifugation. The costliest part of the protein purification is the process resulting in the target enzyme which requires separation from a complex protein mixture containing the own proteins of the host cells as well. When the target enzyme is thermostable, heat shock is a cheap and easy way to get rid of the non-thermostable proteins [54].

To date, a lot of different techniques have been developed for protein separation: e.g., size-exclusion, hydrophobic and affinity chromatography or fractional precipitation [71]. When recombinant protein production is applied, the use of affinity techniques is very common because affinity tags are easy to fuse to the protein by using a suitable vector. Due to the selective interaction between the tag and the affinity function of the chromatographic stationary phase, a high degree of purification can be achieved in a single operation [64]. After taking advantage of the affinity tag, it can be eliminated easily with a chosen site-specific protease if the protein of interest is expressed with the proper protease cleavage site [72]. To find the most suitable tag, the size and nature of the tag, the costs and circumstances of the binding, elution and tag removal should be considered. In many cases, enzymes have only limited stability in the elution buffer, therefore the enzyme isolated after removal of the impurities should be dialyzed against a suitable buffer, followed by concentration and storage for further using.

The purification and polishing steps often require sophisticated apparatus and chromatographic stationary phases. The waste-water generated during the dialysis or stationary phase regeneration steps can also increase the environmental load of the purification process. Therefore, it is highly beneficial if the purification process can be simplified on the way to get the final biocatalyst.

### 1.2. Improved Immobilization Techniques for Biocatalysts: Methods, Pros and Cons

There are several shortcut possibilities from the cell or enzyme production until reaching the final functional form of the biocatalyst (Figure 1). Choosing the right technique of immobilization depends on the desired use of the biocatalyst, considering the activity, selectivity, stability and economic points of view.

The first strategy, when the whole cells are to be immobilized, is applicable for strains isolated from nature or for microbial whole cells hosting expressed wild type or mutant enzymes. In this case, the only step of the downstream processing is harvesting the whole cells after fermentation usually by centrifugation. It is a relatively simple and inexpensive method, which is advantageous if the enzyme is sensitive to environmental effects and/or unstable in extracellular media. Importantly, the whole cells contain usually the cofactors and further enzymes aiding cofactor regeneration, but if there is no substrate for the regeneration of the cofactor, feeding the appropriate cofactors in small concentrations may be necessary for effective long-term operation [32]. Whole-cell immobilization is required when the target enzyme is a cell surface display protein. In this case, it is advantageous that the substrate does not have to diffuse into the cell, because the enzyme is located on the outer surface of the cell [73,74]. However, a disadvantage of the immobilized whole-cell biocatalysts that they are multienzyme systems in which undesired side reactions can occur catalyzed by the other enzymes present in addition to the desired enzyme. In this case, the target enzyme should be separated from the further disturbing enzymes.

The next possibility is immobilization of the cell lysate containing proteins and enzymes separated from the cell debris with centrifugation after cell lysis. Typically, similar immobilization techniques can be used for this protein mixture which are also applicable for a purified enzyme. Immobilization at this phase of downstream process is advantageous for proteins which would not tolerate further purification steps such as chromatography, dialysis or concentration. Another case when the immobilization can take place after the lysis is when the high concentration of the target enzyme allows to omit further purification.

If a partially purified or a homogenous enzyme is needed, further purification steps have to be accomplished. The numbers and type of these depends on the enzyme. If the enzyme is thermostable some of the host proteins can be removed with heat shock [54] and centrifugation. Recombinant production of proteins allows fusing various affinity tags to the target enzyme and using the most efficient affinity chromatographic methods during the downstream processes. For example, strep-tags, S-tag, calmodulin-binding peptide, cellulose- or chitin-binding domains are used in combination with an appropriate chromatographic stationary phase with an immobilized ligand with high affinity and selectivity towards the tag. Affinity chromatography usually ensure a very selective binding and high purity in one chromatographic step [64,75]. One of the most often used tags nowadays is the His-tag comprising of an oligopeptide sequence of several histidines. The His-tag is suitable for metal ion affinity chromatography due to the strong chelating ability towards various metal ions. The following subchapters go into more details of the above-mentioned immobilization techniques enabling shortcuts from the fermentation to a homogenous form of the target enzyme.

#### 1.2.1. Whole-Cell Immobilization Strategies

To immobilize whole cells, entrapment techniques are widely used [76,77]. In this method the cells are retained within a polymer matrix, like capsules, nanofibers or sol-gel matrices, in a way enabling substrate access to and product egress from the cells.

Alginate is a very popular biopolymer for cell entrapment [78]. It is an affordable raw material and easily forms alginate gel matrices in the presence of calcium or other multivalent metal ions. Agar, agarose, chitosan, carrageenans are also suitable polysaccharides for whole-cell immobilization [76,79]. Carrageenans also form gel when added to a solution of calcium salt. By the aid of solvent evaporation, changing pH or using cross-linkers, membranes, films or gels can be formed from chitin, chitosan or gelatin [80]. Other organic and inorganic polymers are also used for the immobilization of cells. The application of poly(vinyl-alcohol) (PVA), acrylic polymers, [76] or polyurethane foams [81] as entrapment matrices also resulted in easy-to-store biocatalysts. Organosilane-based sol-gel matrices offer a general and gentle way for cell entrapment as well [82,83,84]. Depending on the sol formation method and the type of the silica precursor, the sol-gel formation may take place in aqueous or organic media catalyzed by acids or bases [32,83,85,86]. The addition of solid silica particles to this sol-gel system could approve the catalyst diffusion properties or mechanical stability [32].

Besides the formation of gels, hydrogels or membranes, some polymers like PVA, polyethylene oxide (PEO), polyvinyl pyrrolidone (PVP), polycaprolactone (PCL) or polylactic acid (PLA) proved to be suitable for cell-entrapment in electrospun nanofibers [87,88,89,90]. Selection of the suitable polymer requires considerations based on the solvent tolerance of the cells and the conditions of the designed biotransformations. There are fully water-soluble polymers, like PVA or PEO, but in many cases organic solvents are needed to mediate the electrospinning process. An important aspect of matrix selection for electrospun fiber cell entrapment is that the formed nanofiber should not dissolve in the medium of the biotransformation.

Immobilization of whole cells on solid supports is less common. Adsorption is based on secondary interactions, such as ionic interactions [91]. To diminish cell leakage, adsorption as first step can be combined with other immobilization techniques like encapsulation, coatings or covalent cross-linking after flocculation [92,93]. In this case the adsorbents and the flocculants have great impact on the produced biocatalyst. The addition of solid support, such as diatomite, bentonite, silica microspheres, could increase mechanical and chemical stability and the biocatalytic activity compared to the cross-linked or entrapped cells without solid support [32,92]. For cross-linking and covalent immobilization onto solid surfaces, glutaraldehyde is widely used [92,94]. It is getting popular to combine these methods and merge the advantageous properties of the different systems creating new hybrid matrices for whole cell or enzyme immobilization [95,96].

#### 1.2.2. Strategies for Immobilization of Cell-Free Forms of Enzymes

Immobilized forms of enzymes have several advantages over their native forms. Usually, immobilized biocatalysts are easy to recycle and the immobilization often improves resistance of the enzyme against environmental impacts (temperature, pH, ionic strength, organic solvents, etc.) [97].

The methods of enzyme immobilization can be categorized in several ways. In one of the, the immobilization protocols can be categorized into two main types, the physical and the chemical immobilizations, based on whether the enzyme was chemically modified or not. In case of physical immobilization, the enzyme structure is not modified chemically. Two subclasses may be distinguished within this type. First, when the enzyme is entrapped (encapsulated) in a polymer matrix. This can be achieved for example by sol-gel or polysaccharide based gels or by electrospinning, similarly, as already mentioned at the whole cell immobilization methods in Section 1.2.1. [18,98,99,100,101]. The second subclass of non-chemical immobilization methods is based on secondary interactions between the enzyme and a solid support. These secondary interactions can be adsorption based on π-π interactions, hydrogen or ionic bonds, or a stronger affinity-based interaction [102,103]. The later method is suitable for orienting the protein during the immobilization, by positioning the tags in an adequate position within the protein sequence of the functional enzyme. Because these methods avoid chemical modification of the enzyme, it is considered that this kind of immobilization keeps the active form of enzyme with higher probability than the methods based on chemical protein modifications. Methods based on physical interactions are useful when covalent immobilization leads to enzyme inactivation (e.g., by reaction of a lysine in the active center or near to it). In case of the entrapped enzymes, the negligible leakage from properly selected polymer matrixes and the enhanced heat and solvent tolerance are advantageous, but the hindrance of substrate and product diffusion can be much higher than with other types of immobilizations. On the other hand, the adsorbed enzyme biocatalysts have more beneficial diffusion properties, but in this case, a serious disadvantage can be the desorption of the enzyme from the surface which can cause a decrease in the activity and the reusability of the biocatalyst.

The chemical immobilization techniques usually mean immobilizations are based on covalent bond formation. Covalent bonding may happen between the enzyme and some kind of carrier or between several enzyme molecules (Cross-Linked Enzyme Aggregates – CLEAs) [45,104]. Serious disadvantages of the CLEAs method are the high enzyme amount required, the usually weak chemical and mechanical resistance of the resulted biocatalyst and the hindered diffusion due to the formed tight nanopores.

Multipoint covalent immobilization onto a solid support leads to stable biocatalysts where the structural, diffusional and mechanical properties can be influenced with the well-chosen support. Abundantly, glutaraldehyde is applied for the formation of covalent bonds [105]. Besides the often used glutaraldehyde, epoxy functioned supports have been used for a long time [106], and activated carboxyl, amino, hydroxyl or thiol groups are also suitable for covalent immobilization [21,104,107,108,109]. The two-step method of the covalent immobilization can be enhanced with heterofunctional epoxy supports [109,110,111,112]. Different functional groups beside the epoxy functions like thiol [113,114], amino- or carboxylic groups [109,111], hydrophobic functions [103,115] and iminodiacetic acid complexed metal ions [116,117,118,119,120] can help the enzyme adsorption and thus the covalent immobilization. Also the pH during the immobilization has a great impact on the formation of stabile multipoint covalent attachment between the enzyme and the support [114,120]. Nowadays, a great variety of supports are used: macroporous polymers [107,109], silica particles in different size and shape [21], magnetic nanoparticles [121] and several type of nanofibers [122] or nanotubes [123] are just some of the most popular ones. Also modification of the surface with inert groups besides the reactive functions has an effect on the biocatalytic activity [103,109].

### 1.3. Immobilization Methods of Isolated Enzymes Simplifying the Downstream Process

From the economic and environmental point of view it is advantageous if the costly and time-consuming enzyme purification processes can be eliminated fully or partially in the course of the biocatalyst production. As already demonstrated, there are well-known methods enabling selective immobilization [124]. However, these selective methods are usually based on affinity interaction between the enzyme and the surface and not on covalent immobilization.

#### 1.3.1. Embedding as Purification Free Process

The immobilization methods based on embedding and presented in Section 1.2.1 – like sol-gel, alginate or chitosan systems or electrospun nanofibers – are suitable for efficient and usually low cell/protein loss immobilization of whole cells or cell lysate without further protein purification.

#### 1.3.2. Affinity-Based Methods

In the late 1970s, Porath and coworkers discovered that proteins containing histidine and cysteine on their surface can coordinate to transition metal ions via the imidazole and thiol groups of these amino acids [125]. They created an agarose gel with iminodiacetic acid complexed to Zn^2+^ and Cu^2+^ ions. This metal ion-charged agarose gel was used successfully to separate human serum proteins. Inspired by this discovery, Hochuli and coworkers used this system for purification of poly-histidine-tagged proteins [126]. Due to the quite rare occurrence of multiple consecutive histidine sequence in Nature, it is a great opportunity to tag proteins and purify them in an easy way. With the aid of modern molecular biology methods and enzyme engineering, it is now very simple to attach six or ten histidines to a recombinant protein enabling coordination of the tagged protein to transition metal ions. The immobilized metal affinity chromatography (IMAC) as a protein purification process became one of the most often used affinity-based methods for purification of recombinant proteins. One of the most popular commercial purification resin is still based on agarose charged with Ni^2+^ ion chelated with nitrilotriacetic acid (NiNTA), which is similar to that used by Porath in 1983 [127]. Despite the popularity of IMAC, during the last four decades a number of further affinity tags, chelating agents and supports were created for the purification of enzymes and other proteins, like antibodies as well (Table 1) [64,65,66].

However, IMAC (and the other listed methods) are applicable not only for protein purification but are also useful for selective and oriented enzyme immobilization [140]. This process, when applied without elution of the bound protein from the support, enables selective immobilization of the biocatalyst in one-step. Because the immobilization in such case is based on reversible interaction between the fixed metal ion and the protein, in case of enzyme inactivation the exhausted enzyme can be eluted from the surface and the support can be recharged with fresh enzyme in a new immobilization cycle. This is advantageous in case of expensive or hard-to-prepare supports.

Although the use of this complex formation for enzyme immobilization is a very straightforward and rapid way to obtain biocatalysts, but sometimes the conditions of the target biotransformation require more stable immobilization. If enzyme leakage occurs during the application (e.g., a chelating agent is present in the reaction, or the pH or ion concentration do not favor the enzyme-metal complex stability) an immobilization method resulting in more durable binding is needed. A good strategy may be to stabilize the complexed enzyme after binding to the IMAC support by other methods. As a further step after IMAC support binding, embedding methods can be considered (such as electrospinning or sol-gel entrapment of the enzyme on the support in an appropriate matrix). Alternatively, covalent bonds can be created after the complexation if there are further reactive functions on the support. Such strategy was applied in a modified, selective CLEA method for the immobilization of His-tagged acid phosphatases [141]. Smart silica nanoparticles with amino functions and metal chelating groups were created enabling selective immobilization of acid phosphatases in a process starting with rapid affinity binding of the target phosphatases followed by cross-linking the enzyme-nanoparticles system with bisepoxides as cross-linkers. The created acid phosphatase biocatalysts showed excellent activity and long-term operation stability compared with the enzyme immobilized by other methods.

In case of conventional covalent immobilization, usually the use of purified enzymes is required because the often used aldehyde- or epoxy-group based binding functions on the support cannot distinguish the amino groups of different proteins. Since the costliest part of the enzyme production is the purification, it is desirable to develop single process methods for simultaneous enrichment and immobilization. Thus it is advantageous to combine the two methods within one process or on one support. When a covalently immobilized biocatalyst is needed in a single operation process without breaking off the procedure between the purification and stabilization steps, the chelating and covalent binding functions must be present in the same time on the support. A selective and stabile process can be achieved by these strategies if a first rapid complexation is followed by a slower covalent binding step [120].

Mateo and coworkers immobilized different enzymes on multifunctional epoxy supports, also on iminodiacetic acid (IDA) and Cu^2+^ modified Eupergit C [109,116]. Later they optimized the epoxy and IDA-metal ion ratio with different reaction times of the Eupergit C and used the modified support for the immobilization of His-tagged glutaryl acylase [120]. It was found that only a small surface metal ion concentration was enough to exert the selective effect. Most of the target proteins adsorbed on the surface when the Co^2+^ concentration was 5 µmol/mL with IDA – epoxy ratio 1 to 7 on the support and the binding of the other proteins was negligible. In this way it was possible to gain immobilized biocatalyst from the crude cell lysate with almost the same biocatalytic activity as from the purified enzyme solution.

There are further examples from the last twenty years for using similar bifunctional supports for one step purification and immobilization [117,118,119,120]. Penicilline G acylase was immobilized on epichlorohydrin, iminodiacetic acid and Cu^2+^ functionalized cellulose membrane [117]. The amount of metal chelating groups was controlled by using different amounts of IDA during the surface modification. In this case, the optimal ratio of epoxy and metal chelating groups were close to 1:1 in the preparations providing the most stable and reusable biocatalyst.

Also magnetic nanoparticles with an epoxy – IDA combination in 13:2 ratio were used for selective immobilization of His-tagged benzaldehyde lyase (BAL) [118]. After the elution of the non-covalently attached proteins with imidazole buffer, a covalently immobilized biocatalyst was obtained directly from the crude cell lysate. The BAL-MNPs biocatalyst was used successfully in the condensation reaction of benzaldehyde to benzoin.

Alcohol dehydrogenase from *Haloferax volcanii* was immobilized on commercially available epoxy support, Sepabeads EC-EP/S, after modification approximately 5% of epoxy groups with IDA [119]. Three transition metal ions – Co^2+^, Ni^2+^, Zn^2+^ – and Ca^2+^ were tested in the immobilization process. The best efficiency was earned with Ni^2+^ modified support.

In all of these examples an epoxy support was partially modified with IDA as a metal chelating agent. In all instances, the mixed epoxy-chelate function supports were more effective than the epoxy supports. Creating the chelating functions directly on an epoxy support is quick and easy, but the fine tuning of the surface is more complicated due to the restricted variability of the required multi-linker modifications. In our work, a similar immobilization technique was used, but fine-tunable surface modification of the carrier system was developed. Both the epoxide moieties and the complexing agent were formed simultaneously during a single surface modification step of an amine functionalized support. This allowed us to vary the properties of the final selective support by choosing and combining different agents of variable chemical structures, lengths and the hydrophilic-hydrophobic properties to form the two functions for chelation and covalent binding. All of these properties have an impact on the immobilization of the enzyme and on its final biocatalytic behavior.

Phenylalanine ammonia-lyase (PAL) from various organisms has a great impact in the production of several unnatural amino acids [142,143], and also get a lot of attention as a possible enzyme substitution treatment for patients suffering from phenylketonuria [144]. Therefore, immobilization of a phenylalanine ammonia-lyase has been selected as the target of the present study. Based on these preliminary considerations, our aim was to prepare mixed epoxy-chelate functionalized magnetic nanoparticles with ethylenediaminetetraacetic acid related chelating function and test them as selective support in a single-step immobilization of PAL.

## 2. Results and Discussion

In this work we studied the advantages of the epoxy-chelate bifunctional supports in the immobilization of phenylalanine ammonia-lyase from *Petroselinum crispum* (*Pc*PAL) expressed in recombinant *E. coli* even from crude cell lysate (Figure 2). Magnetic nanoparticles were selected as support because of their easy separability, high surface area and simple surface modification [145]. Silica shell-coated magnetic nanoparticles were etched with aminopropyl functions and further modified with mixtures of ethylenediaminetetraacetic dianhydride (EDa) and various multiepoxide compounds in different ratio. With this strategy it was possible to test different multiepoxides to form the covalent binding function and also the ratio of chelating and covalent binding sites could be modulated.

Our previous experiences with the IMAC purification of *Pc*PAL indicated that the strongest and most selective binding could be achieved by cobalt(II) ion charged supports [unpublished data]. Therefore, no further metal ion tests were performed in this study, and cobalt(II) was used solely as a chelating metal ion. The formed *Pc*PAL biocatalysts were tested in ammonia eliminating reaction from l-phenylalanine [146].

### 2.1. Optimizing the Epoxy-Chelate Ratio

First, to determine the proper amount of multiepoxide and ethylenediaminetetraacetic dianhydride (EDa) as admixture in the amine-modification reaction, the amount of the amino groups at the surface of aminopropyl-etched MNPs had to be measured. The amino group content on the surface of the MNPs was quantified by ninhydrin colorimetric assay [147] (see Appendix A) and was found to be 360 ± 18 µmol × g^−1^.

Because the dianhydride was the more reactive component in the surface modification reaction, being also sensitive to hydrolysis, the surface amine-modification reactions were carried out in dry dimethylformamide. In the first series of experiments, neopentylglycol diglycidyl ether (NPDGE) was used as a bisepoxide agent. In addition to the pure chelating and pure epoxy MNPs supports, nine further MNPs supports prepared with different ratio of NPDGE and EDa. The functionalization mixtures were investigated for the immobilization of *Pc*PAL leading to biocatalysts with different immobilization yields and biocatalytic activities (Figure 3). After the overnight shaking with the cell lysate containing *Pc*PAL, the samples were washed with imidazole solution (500 mM) in order to elute the weakly complexed His-tagged enzymes and other histidine contained proteins.

It is clearly shown that for the most effective immobilization the presence of both the complexing and the covalent binding groups were required (Figure 3). In the absence of epoxy groups, the enzymes attached to the surface due to the affinity tag chelation and further ionic interactions, and as expected, the *Pc*PAL-binding MNPs had reasonable biocatalytic activity. Because the expression level of *Pc*PAL was quite low in our present case, the cell lysate contained high amounts of other proteins besides the target enzyme. Therefore, it is not surprising that the epoxy-containing MNPs without the ability to distinguish between the proteins gave only a poor specific *Pc*PAL activity. The best specific activity could be achieved when the EDa and NPDGE were used in 1:10 ratio in the amine-modification reaction mixture. When the ratio of metal chelating groups to NPDGE decreased below 1:10, the specific biocatalytic activity was reduced due to limited capacity to select the target enzyme from the protein mixture during the immobilization. With MNPs of the best ratio, 33× better specific activity of the immobilized *Pc*PAL biocatalyst could be achieved as with MNPs containing solely epoxy functions on the surface. 

### 2.2. Comparison of the Biocatalytic Activity of Simple Epoxy – and Mixed Epoxide/EDa Functionalized MNPs

To demonstrate the enhanced effectivity of this immobilization method, *Pc*PAL immobilization with the optimized mixed epoxide/chelator functionalized MNPs was compared to the epoxy-functionalized MNPs from purified *Pc*PAL solution and from a crude lysate after *Pc*PAL expression (Table 2).

Immobilization of *Pc*PAL from crude lysate with the surface optimized support resulted in almost as high activity as immobilization from the purified enzyme on the optimized mixed epoxy-chelate MNPs or on the epoxy functionalized MNPs. It means that by using the optimized mixed epoxy-chelate MNPs for immobilization of *Pc*PAL the protein purification process could be skipped by a single process operation leading to the same specific activity as achievable with simple supports only from purified enzyme solution.

In our present case, *Pc*PAL exhibited a low level of expression in *E. coli*, usually resulting in 3–5% active enzyme content in the total proteins of the cell lysate. The 33 times better specific activity of *Pc*PAL immobilized on mixed MNPs-epoxy-chelate as compared to MNPs-epoxy clearly indicated the benefits of this selective enzyme immobilization method when enzymes are produced at low expression levels. However, to provide guidance in which cases this method is worth using, we investigated the immobilization of *Pc*PAL with the optimal epoxy-chelate MNPs compared to the epoxy only MNPs as support for immobilizations from protein mixtures of six different target protein contents. For accurate modeling, known amounts of purified enzyme in lysis buffer were added to cell lysate to mimic mixtures representing different expression levels. In the final mixtures the concentration of the target enzyme was set to the same value and immobilizations were performed with the MNPs-NPDGE/EDa-10 and the MNPs-NPDGE supports (Table 3).

The results clearly indicated the benefits of the optimized MNPs-NPDGE/EDa-10 for selective single operation immobilization of *Pc*PAL not just in expression levels below 10 *m*/*m* % target enzyme content (>13-fold enhancement in specific activity compared to the non-selective MNPs support) but even in case of 50 *m*/*m* % target protein concentration (1.6-fold enhancement).

### 2.3. Effect of the Nature of Multiepoxide Agents on the MNPs for the Immobilization of PcPAL

Another advantage of the post modification of amine functionalized magnetic nanoparticles is the variability of the applicable multiepoxides. Because the amino MNPs do not contain the epoxy groups inherently, they have to be created together with the complexing functions. In this study, ten multiepoxides were used for the surface functionalization of MNPs with aminopropyl functions at the best epoxy-EDa ratio of 10:1 (Figure 4).

After surface activation with the ten different multiepoxides and EDa in 10:1 ratio, the resulted epoxy-chelate MNPs were tested as supports in the selective immobilization of *Pc*PAL (Figure 4). There was no significant difference between specific biocatalytic activities of seven bisepoxides and two trisepoxides (*U_B_* 10 to 14 U × g^−1^), but the mixture formed from the bulkiest trisepoxide (THPMTGE) resulted in a significantly higher specific activity (24.7 U × g^−1^). However, the relative residual enzyme activity compared to the native enzyme activity was significantly lower with the MNPs-THPMTGE/EDa-10 based biocatalyst than with the MNPs-NPDGE/EDa-10 support. It was observed that epoxides with a shorter and substituted alkyl chain (GDGE/NPDGE/GTGE/TMPTGE) or epoxides with aromatic rings (BADGE/THPMTGE) as covalent linkers resulted higher specific biocatalytic activities and activity yields that obtained with epoxides containing longer mobile chains or aliphatic ring (PDGE/BDGE/CDGE/HDGE).

### 2.4. Operational Stability of the Immobilized PcPAL Biocatalysts

Besides of the better mechanical and chemical properties, one of the biggest benefits of using immobilized biocatalysts is the reusability. Two immobilized *Pc*PAL biocatalysts were chosen for repeated batch tests (the immobilization leading to the *Pc*PAL-MNPs-NPDGE/EDa-10 and the *Pc*PAL-MNPs-THPMTGE/EDa-10 was scaled up thirty-fold). The selectivity of these two supports for the target protein *Pc*PAL were investigated by SDS-PAGE. The covalently attached protein composition was estimated by SDS PAGE analysis of the adsorbed protein onto the MNPs surface after a 1 h immobilization time with the cell lysate (Appendix A). Based on the SDS-PAGE analysis, most of the adsorbed proteins were the target enzyme *Pc*PAL indicating that the bifunctional epoxy-chelate supports had an excellent selectivity for the His-tagged enzyme.

The biocatalysts were tested by five repetitive cycles in the ammonia elimination of racemic phenylalanine (d,l-PHE) in a traditional buffer system (TRIS, 100 mM, pH 8.8). The conversions and enantiomeric excess values achieved by the immobilized biocatalyst are presented in Table 4. Although the activity of both biocatalysts decreased cycle-by-cycle, after five batch reaction the catalysts retained 90% of their initial activities.

The reusability of these biocatalysts were tested also under more demanding conditions in the opposite reaction direction to catalyze ammonia addition onto *trans*-cinnamic acid (*trans*-CA) producing l-phenylalanine. The activity of the *Pc*PAL-MNPs-THPMTGE/EDa-10 was barely diminished during the recycling even of the negative effect of the long-term presence of high ammonia concentration [123]. Both biocatalysts produced l-phenylalanine at high conversion rate with an excellent enantiomeric excess even after five repeated batch reactions (Table 5).

## 3. Materials and Methods

### 3.1. Materials and Analysis

The magnetic nanoparticles coated with aminopropyl functional groups–MagnAmine_AP_–were kindly donated by SynBiocat Ltd. (Budapest, Hungary). Bisepoxides 1,4-butanediol diglycidyl ether (BDGE), neopentylglycol diglycidyl ether (NPDGE), 1,6-hexanediol diglycidyl ether (HDGE), 1,4-cyclohexanedimethanol diglycidyl ether (CHDGE), polyoxypropyleneglycol diglycidyl ether (PEDGE), trimethylolpropan triglycidyl ether (TMPTGE) and glycerol triglycidyl ether (GTGE) were the products of Ipox Chemicals Ltd. (Budapest, Hungary). Glycerol diglycidyl ether (GDGE), Bisphenol A diglycidyl ether (BADGE), tris(4-hydroxyphenyl)methane triglycidyl ether (THPMTGE), ethylenediaminetetraacetic acid (EDTA), potassium chloride, sodium chloride, tris(hydroxymethyl)aminomethane (TRIS), cobalt(II) acetate tetrahydrate and dialysis tubing cellulose membrane with a 14 kDa cut off were purchased from Sigma-Aldrich (Saint Louis, MO, USA). Imidazole, 4-(2-hydroxyethyl)-1-piperazineethanesulfonic acid (HEPES), Bradford reagent, l-phenylalanine (l-Phe) and trans-cinnamic acid were purchased from Alfa Aesar Europe (Karlsruhe, Germany). Pyridine, acetic anhydride and all solvents were purchased from Merck KGaA (Darmstadt, Germany). The technical grade solvents were dried and/or freshly distilled prior to use and dimethylformamide was dried over 4 Å molecular sieves. The UV-VIS measurements were carried out in a Genesys 2 type spectrophotometer (Thermo Fisher Scientific Inc., Waltham, MA, USA).

### 3.2. Expression and Purification of PcPAL

Expression of recombinant phenylalanine ammonia-lyase from parsley (*Pc*PAL) carrying an *N*-terminal His_10_-tag in *E. coli* Rosetta^TM^ host was carried out at Fermentia Ltd. according to the method described by Dima et al. [146]. To determine the target *Pc*PAL concentration, a small amount of cell lysate was purified on Ni-Sepharose according to the protocols of the manufacturer. For complete recovery of the target protein elution was performed with 500 mM imidazole (in 50 mM HEPES, 30 mM KCl, pH 7.5). The eluted *Pc*PAL was dialyzed against lysis buffer (50 mM TRIS, 150 mM NaCl, pH 7.5) and the dialyzed *Pc*PAL solution was stored at −20 °C. The total protein concentration in the cell lysate was 25.6 ± 2.01 g × mL^−1^ containing the target *Pc*PAL in 0.99 ± 0.13 g × mL^−1^ concentration determined by Bradford method (for the measurement description see Appendix A.).

### 3.3. Surface Treatment of Aminopropylsilane-Coated Magnetic Nanoparticles with EDa and Multiepoxides at Different Ratios

In a dried 4 mL screw cap glass vial was sonicated a mixture of MNPs (50.0 mg; 18 µmol amine function), PEG 400 (25.0 mg of polyethylene glycol with an average molecular weight of 400 g × mol^−1^) in *N,N*-dimethylformamide (DMF, 750 µL) for 10 min. Then variable amounts of a multiepoxide solution (100 µmol × mL^−1^ in DMF), EDa solution (40 µmol × mL^−1^ in DMF) and *N*-ethyl-*N,N*-diisopropylamine (D*i*PEA) solution (60 µmol × mL^−1^ in DMF, 2 equivalents to the EDa) were added to the suspension. The total amounts of EDa and the multiepoxide were two equivalents related to the amino group content of MNPs (36 µmol) but in different molar ratios (EDa-multiepoxide: 1–0; 1–1; 1–5; 1–10; 1–25; 1–50; 1–75; 1–100; 1–150; 1–200; 0–1). The total reaction volume was filled up to 1800 µL by addition of the required amount of DMF. The reaction mixture was shaken at 600 rpm for 24 h at 60 °C, then 50 µL distilled water was added to the mixture which was shaken for further 1 h at 60 °C. Than the surface-treated MNPs were separated with a neodymium magnet from the reaction medium and were washed with acetonitrile (2 × 1.0 mL) and 2-propanol (1.0 mL), dried in a VDL 23 vacuum drying chamber (Binder GmbH, Tuttlingen, Germany) at room temperature overnight. Ethylenediaminetetraacetic dianhydride was synthesize by us (see Appendix A.).

### 3.4. Metal Ion Complexation of the Surface-Treated MNPs and Immobilization of PcPAL

The metal ion complexation of the surface-treated MNPs and immobilization of phenylalanine ammonia-lyase on them were performed in 4 mL screw cap glass vials. The bifunctional metal chelate-epoxy MNPs (5 mg) in distilled water (500 µL) were sonicated for 10 min. After addition of cobalt(II) acetate solution (100 mM, 500 µL), the resulted mixture was shaken for 30 min, followed by separation of metal-charged bifunctional MNPs which were washed with distilled water (3× 1 mL) and lysis buffer (1 mL, 50 mM TRIS, 150 mM NaCl, pH 7.5).

The crude cell lysate or the purified enzyme solution was defrosted and centrifuged (3500 rpm, 2 min, 15 °C). The supernatant protein solution (1 mL) was added to the actual metal-charged bifunctional MNPs samples. The resulted suspensions were shaken for 20 h at room temperature. After the first stage of immobilization, the proteins adsorbed nonspecifically were removed by washing first with low salt buffer (1 mL, 30 mM KCl, 50 mM HEPES, pH 7.5), and then with high salt buffer (1 mL, 300 mM KCl, 50 mM HEPES, pH 7.5) solutions. Next, the non-covalently bonded His-tagged proteins were removed from the cobalt(II)-charged MNPs by washing with an imidazole solution (1 mL, 500 mM in low salt buffer). Finally, the immobilized *Pc*PAL biocatalysts were washed with TRIS buffer (3× 1 mL, 100 mM, pH 8.8) and were tested directly in the ammonia elimination reaction. In the case of two selected surface modified supports (MNPs-NPDGE/EDTA DA-10 and MNPs-THPMTGE/EDTA DA-10), a scale-up of the immobilization was performed (see Appendix A.).

### 3.5. Activity of the Immobilized PcPAL Biocatalysts in the Ammonia Elimination of l-Phenylalanine

In 4 mL screw cap glass vial l-phenylalanine solution (2 mL, 10 mM l-PHE in 100 mM TRIS, pH 8.8) was added to the *Pc*PAL-MNPs biocatalyst (5 mg) and the resulted suspension was shaken at 600 rpm, 30 °C. After 30 min and 60 min samples (50 µL, each) were taken and diluted to 1 mL with distillated water and the absorbance of the samples was measured at 290 nm.

Conversion of l-phenylalanine to *trans*-cinnamic acid was determined from the measured concentration of cinnamic acid at 290 nm by using the Lambert-Beer equation. To characterize the productivity of the *Pc*PAL-MNPs biocatalysts, the specific biocatalytic activity was calculated using the equation *U*_B_ = *n*_P_/(*t* × *m*_B_) (where *n*_P_ [μmol] is the amount of the product, *t* [min] is the reaction time and *m*_B_ [g] is the mass of the applied biocatalyst; in case of the cell lysate *m*_B_ [g] was calculated from the total protein concentration). The specific enzyme activity was calculated using the equation *U*_E_ = *U*_B_/*m*_E_ (where *m*_E_ [g] is the mass of the *Pc*PAL in the applied biocatalyst; in case of the cell lysate*m*_E_ [g] was calculated from the determined *Pc*PAL concentration of the lysate). Activity yield was calculated using the equation *Y*_A_ = 100 × *U*_E_(cat)/*U*_E_(lys) (where *U*_E_(cat) is the *U*_E_ of the biocatalyst and *U*_E_(lys) is the *U*_E_ of the cell lysate). To determine *m*_E_, the activity of the crude cell lysate was measured before and after the immobilization process as well as the imidazole elution fractions during the washing steps (for representing activity measurement see the Appendix A.).

## 4. Conclusions

In this study an effective immobilization strategy that does not require preliminary protein purification steps has been developed by the creation of optimized compositions of mixed epoxy-chelate functionalized magnetic nanoparticles. Our results with the selective immobilization of parsley phenylalanine ammonia-lyase fused to a poly-histidine tag proved that this method is especially useful for the immobilization of a target enzyme from a crude lysate without the need of purification even if the enzyme is expressed at a low level. We believe, that our flexible method enabling the variation and fine-tuning of the covalent linker offers a good opportunity for effective and selective one-step purification and immobilization of further sensitive, or hard-to-purify enzymes decorated with a proper metal binding affinity tag.

## Figures and Tables

**Figure 1 molecules-24-04146-f001:**
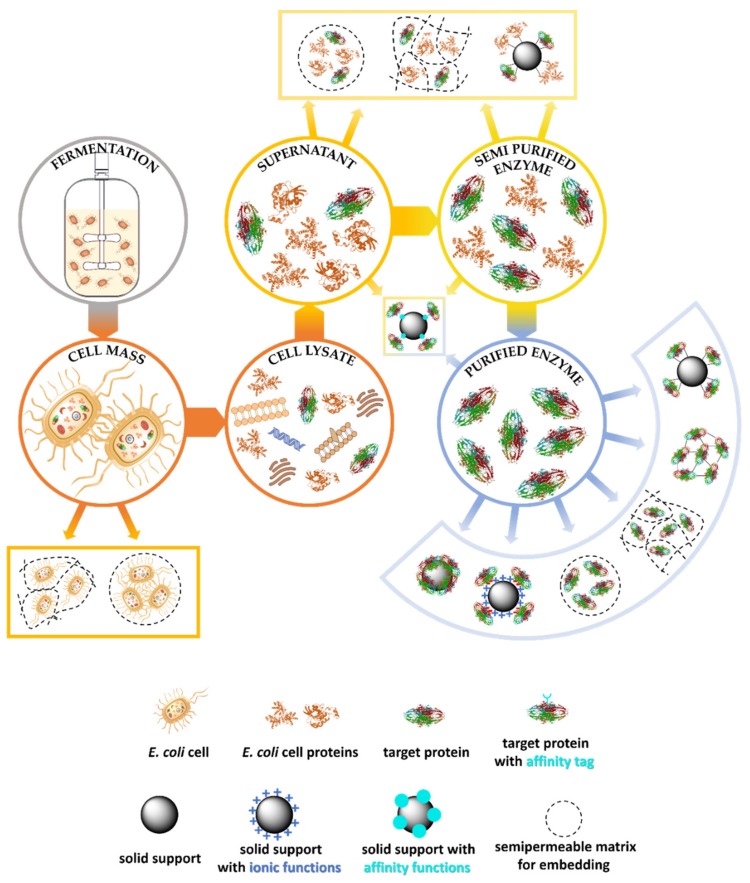
Possible shortcuts of the downstream process during immobilized biocatalyst production.

**Figure 2 molecules-24-04146-f002:**
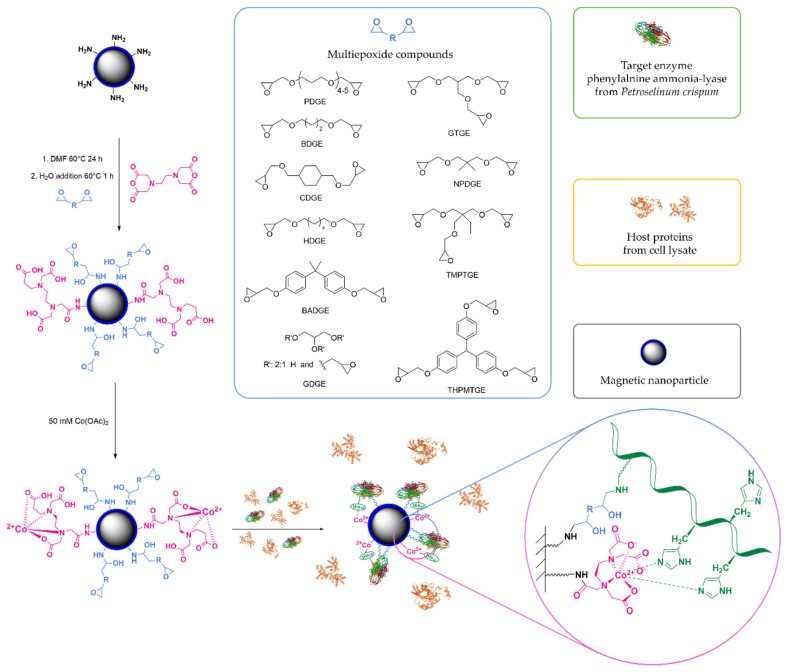
Surface modification of magnetic nanoparticles and immobilization of phenylalanine ammonia-lyase from *Petroselinum crispum* (*Pc*PAL) onto mixed epoxy-chelate magnetic nanoparticles from cell lysate.

**Figure 3 molecules-24-04146-f003:**
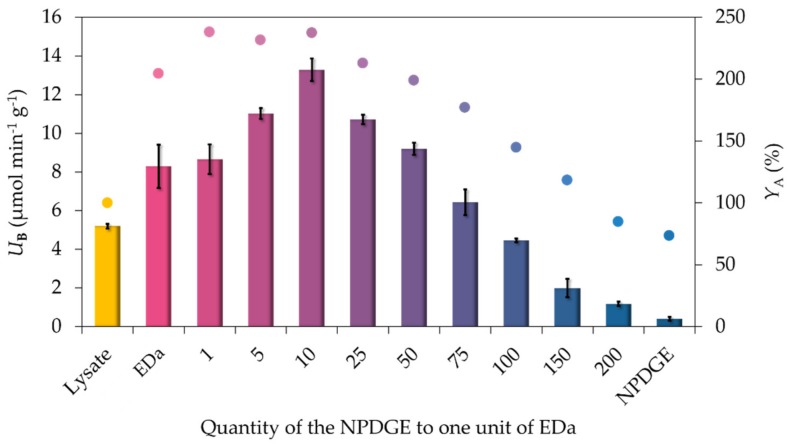
Effect of the bisepoxide to ethylenediaminetetraacetic dianhydride ratio in surface functionalization on the activity of the immobilized *Pc*PAL biocatalyst. Lysate: crude protein mixture without immobilization; EDa: only EDa modified support; NPDGE: only NPDGE modified support; numbers: quantity of NPDGE to one unit of EDa during the surface modification. Colored bars: specific biocatalytic activity (*U*_B_), colored dots: activity yield (*Y*_A_); for the non-immobilized cell-lysate *U*_B_ = *U*_E_. The measurements were performed in three replicates; the standard deviation of activity yield values was always below 5%. Experiments were performed as described in Section 3.5.

**Figure 4 molecules-24-04146-f004:**
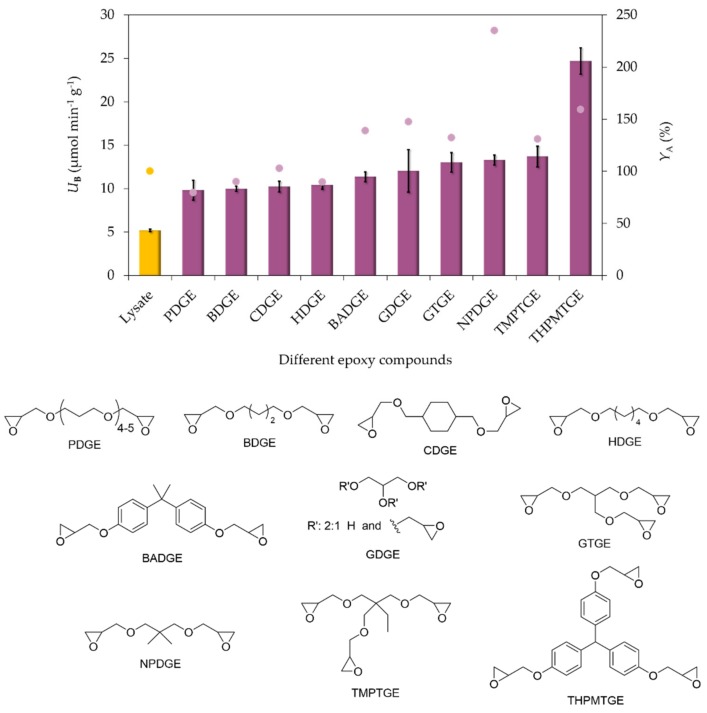
Effect of the nature of multiepoxide linkers on the activity and immobilization efficiency of *Pc*PAL biocatalysts. Bars: specific biocatalytic activity (*U*_B_), dots: activity yield (*Y*_A_); for the non-immobilized cell-lysate *U*_B_ = *U*_E_. The measurements were performed in three replicates; the standard deviation of activity yield values was always below 5%. Experiments were performed as described in Section 3.5.

**Table 1 molecules-24-04146-t001:** Affinity tags for protein purification (protein binding).

Tag Name	Length	Binding Matrix
Poly Arg-tag [128]	5–6 arginine	Cation-exchange resin
Poly His-tag [126]	6–10 histidine	Immobilized metal-coated support
FLAG [129]	8 amino acids	Anti-FLAG MAbs
Strep-tag II [130]	8 amino acids	Modified streptavidin
Calmodulin-binding peptide [131]	26 amino acids	Calmodulin
Cellulose-binding domains [132]	27–129 amino acids	Cellulose
SBP [133]	38 amino acids	Streptavidin
Chitin-binding domain [134]	51 amino acids	Chitin
Si-Tag (L2, Z_basic2_ proteins) [135,136]	58 and 273 amino acids	Silica surface
Glutathione S-transferase [137]	211 amino acids	Glutathione
HaloTag [138]	237 amino acids	HaloTag ligands
Maltose-binding protein [139]	396 amino acids	Cross-linked amylose

**Table 2 molecules-24-04146-t002:** Comparison of *Pc*PAL immobilization from crude lysate or from pure enzyme solution using epoxy and optimized mixed epoxy-chelate MNPs as support [*U*_B_ (μmol × min^−1^ × g^−1^)].

*Pc*PAL Solution	*Pc*PAL-MNPs-NPDGE	*Pc*PAL-MNPs-NPDGE/EDa-10
Crude lysate	0.4 ± 0.1	13.7 ± 0.6
Purified *Pc*PAL	14.2 ± 0.5	14.1 ± 0.6

**Table 3 molecules-24-04146-t003:** Comparison of the efficiency of *Pc*PAL immobilization [*U*_B_ (μmol × min^−1^ × g^−1^)] with epoxy and optimized epoxy-chelate supports from protein mixtures of different target enzyme content.

Target Protein Concentration	*Pc*PAL-MNPs-NPDGE	*Pc*PAL-MNPs-NPDGE/EDa-10	*U*_B_^NPDGE/EDa−10^/*U*_B_^NPDGE^
5	0.5	12.3	24.1
10	1.0	12.6	13.2
25	3.3	13.2	4.0
50	8.3	13.4	1.6
75	11.9	13.6	1.2
90	12.4	13.6	1.1

**Table 4 molecules-24-04146-t004:**
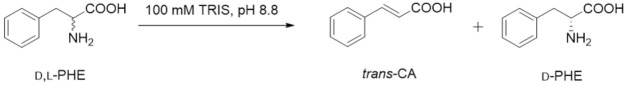
Conversions and enantiomeric excess values of the operational stability tests of immobilized *Pc*PAL biocatalysts in the ammonia elimination reaction of d,l-phenylalanine. All the measurements were performed in triplicate, and the standard deviations were below 5%. For experimental conditions see Appendix A

Reaction Cycle	*Pc*PAL-MNPs-NPDGE/EDa-10	*Pc*PAL-MNPs-THPMTGE/EDa-10
*c* [%]	*ee_d-PHE_* [%]	*c* [%]	*ee_d-PHE_* [%]
1	49	82	52	93
2	48	78	51	92
3	45	71	48	91
4	43	67	47	91
5	43	64	47	89

**Table 5 molecules-24-04146-t005:**
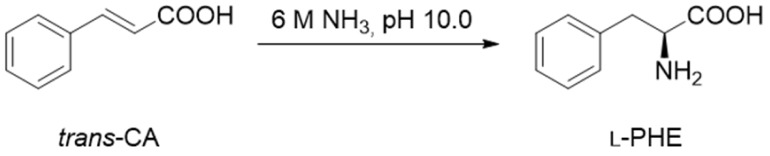
Conversions and enantiomeric excess values of the operational stability tests of immobilized *Pc*PAL biocatalysts in the ammonia addition onto *trans*-cinnamic acid. All the measurements were performed in triplicate, and the standard deviations were below 5%. For experimental conditions see Appendix A.

Reaction cycle	*Pc*PAL-MNPs-NPDGE/EDa-10		*Pc*PAL-MNPs-THPMTGE/EDa-10	
	*c* [%]	*ee_l-PHE_* [%]	*c* [%]	*ee_l-PHE_* [%]
1	85	»99	86	»99
2	86	»99	87	»99
3	85	»99	86	»99
4	82	»99	86	»99
5	70	»99	83	»99

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
