# Peer review of "“Fishing and Hunting”—Selective Immobilization of a Recombinant Phenylalanine Ammonia-Lyase from Fermentation Media"

_molecules, 2019, doi:10.3390/molecules24224146_

Round 1

Reviewer 1 Report

The manuscript “Fishing and hunting”– Selective immobilization of a recombinant phenylalanine ammonia-lyase from fermentation media" describes an enzyme immobilization method on magnetic nanoparticles, employing both imac and covalent binding with epoxide reagents. The paper is very informative and interesting. The format of the paper, merging a review and a research article is also valuable. However, I have several concerns:

Major points
1) Lines 292-317. The authors report a number of previous reports which have employed similar immobilization techniques. In what aspect is this work innovative? Is it less expensive, more rapid, etc?
2) One of the main reasons for enzyme immobilization is to increase stability and reusability (recovery). However, both of these aspects have not been investigated. The paper would be strengthened by an experiment of enzyme recycling.
3) A figure illustrating the protocol for the preparation of the immobilized enzyme would be very helpful
4) Is it possible that some non-his-tagged protein was covalently bound to MNP? how was this checked?

Minor points
- line 110 "restriction enzyme". This term is commonly referred to endonucleases. For the sake of clarity, I suggest using "site-specific protease" or something similar
- most of the times the authors list several approaches or materials without a brief evaluation (e.g lines 174-177, 227-229)
- line 190-192; please provide more detail
- line 197: "...In one way.."; what way?
- line 221-222 "..rough structural properties..". Please be more specific
- line 259 IMAC and following abbreviations (e.g. line 293); please define them
- line 351; introduce here the multi-epoxide reagents (Figure 4)
- line 421-422. Not clear here what is the "activity yield". What does this parameter indicate? (it is indicated in the methods but here the significance of the parameter should be reported).

Weird sentences, requiring rephrasing. A general revision is also suggested.
- 78 "fag"
- lines 134-136
- lines 142-143
- line 177; "..can approve the catalyst.."
- lines 218-219 "The chemical immobilization techniques usually mean immobilizations are based on covalent bond formation"
- line 252: Why "Although"?

Author Response

Answers of the questions of Reviewer 1

Manuscript ID: molecules-615889

Article title: “Fishing and hunting”– Selective immobilization of a recombinant phenylalanine ammonia-lyase from fermentation media

Major points

1) Lines 292-317. The authors report a number of previous reports which have employed similar immobilization techniques. In what aspect is this work innovative? Is it less expensive, more rapid, etc?

Response: The innovation of our method based on the rational surface modification of magnetic nanocarriers for selective enzyme immobilization. The great variability in the used complexing agent (every complexing agent which can form an anhydride) and multiepoxide compounds provide fine tunable surface properties for the simultaneous complexation and covalent binding of a his-tagged recombinant enzyme. This strategy can simplify the enzyme purification and immobilization steps resulting in cost-effective downstream processes. In addition, the surface building protocols elaborated on magnetic nanoparticles reported here can be a general approach for other selective immobilization issues. A brief summary related the above mentioned lines was insert in to the section given by the reviewer.

2) One of the main reasons for enzyme immobilization is to increase stability and reusability (recovery). However, both of these aspects have not been investigated. The paper would be strengthened by an experiment of enzyme recycling.

Response: In addition reusability test were performed in five recycle round with two biocatalysts (PcPAL-MNPs-NPDGE-EDa-10 and PcPAL-MNPs-THPMTGE-EDa-10) in the ammonia elimination reaction of d,l-phenylalanine and also in the ammonia addition reaction of trans-cinnamic acid. All the reactions were performed in 3 repetitions. The results are included in the publication in a new subsection: 2.4 Operational stability of the immobilized PcPAL biocatalysts.

3) A figure illustrating the protocol for the preparation of the immobilized enzyme would be very helpful.

Response: A new scheme was added to the publication about the surface modification procedure instead of Figure 2.

4) Is it possible that some non-his-tagged protein was covalently bound to MNP? how was this checked?

Response: The simplest and more general way to investigate the composition of a protein mixture is the SDS-PAGE analysis. While the covalently attached proteins cannot be investigated in this way (Mateo et al. Biotech&Bioeng, 2001, 76, 269-276.) we estimated it with the adsorbed proteins from the crude fermentation media on the MNPs surface after one hour immobilization time. Because the first step of the covalent immobilization is the adsorption of the proteins on the surface, this is a good prediction for presence of the covalently attached proteins. These results were attached in the Supporting information at Section 3.2.

Minor points

1) Line 110 "restriction enzyme". This term is commonly referred to endonucleases. For the sake of clarity, I suggest using "site-specific protease" or something similar.

Response: Thank you for the reviewer’s comment. The sentence was corrected.

2) Most of the times the authors list several approaches or materials without a brief evaluation (e.g lines 174-177, 227-229).

Response: Thank you for the reviewer’s comment. The authors endeavored to meet the requirements of featured article and give a summary about the purification free immobilization techniques. Most of these techniques (whole cell immobilization, protein mixture immobilizations in polymer matrixes) deserved a whole book or book chapter, and hard to find the balance between the review like summary and the experimental results. For this reason, we decided to detail only the directly related publications when researchers used epoxy-chelate systems.

3) Line 190-192; please provide more detail.

Response: Thank you for the reviewer’s comment. Some more details were described about the combined adsorption-crosslinking and adsorption-entrapped methods.

4) Line 197: "...In one way.."; what way?

Response: Thank you for the reviewer’s observation. The sentence was clarified as follows: “One of them when the immobilization protocols were categorized into two main types, the physical and the chemical immobilizations, based on whether made chemical modification on the enzyme or not.”

Instead of:” In one way, the immobilization methods can be categorized into two main types, the physical and the chemical immobilizations.”

5) Line 221-222 "..rough structural properties..". Please be more specific.

Response: Thank you for the reviewer’s observation. The sentence was clarified as follows: “A serious disadvantage of the CLEAs method is that high enzyme amount is required, and the resulting biocatalyst usually has poor diffusion properties in the formed too tight nanopores, weak chemical and mechanical resistance.”

Instead of: “A serious disadvantage of the CLEA method is the high enzyme amount that have to be used, and the rough structural properties of the biocatalyst.”

6) Line 259 IMAC and following abbreviations (e.g. line 293); please define them.

Response: We introduced the abbreviation IMAC at the first mentioning of immobilized metal ion affinity chromatography in line 252. Also for the IDA in line 294.

7) Line 351; introduce here the multi-epoxide reagents (Figure 4).

Response: In our opinion, the summarizing figure about the several multiepoxides it is better to be placed right before the experimental results connected these multiepoxide compounds modified biocatalysts. However we introduced them in the redesigned Figure 2.

8) Line 421-422. Not clear here what is the "activity yield". What does this parameter indicate? (it is indicated in the methods but here the significance of the parameter should be reported).

Response: Thank you for the reviewer’s comment. It was defined in the main text what the activity yield means.

Reviewer 2 Report

“Fishing and hunting”– Selective immobilization of a recombinant phenylalanine ammonia-lyase from fermentation media

Comments for the Authors

This article describes the immobilization of phenylalanine ammonia-lyase (PAL) onto magnetic nanoparticles (MNPs) using a bifunctional approach. Bifunctional MNPs were prepared by reacting aminopropyl functionalized MNPs with different molar ratios of the metal chelating EDTA dianhydride and multiepoxides. These bifunctional MNPs could thus be used to enrich his-tagged proteins directly form cell lysates of his-tagged proteins using IMAC and could then be selectively and covalently attached to the MNP. The authors showed how various ratios of EDTA DA and different multiepoxides changed the resulting activity of recombinantly expressed PAL (with an attached (His)10 tag) directly from cell lysate.

This is a well-written article with a very complete introduction and whose results mostly (without the complete kinetic data) justify the conclusions. The use of a bifunctional MNP to prepare covalently attached enzymes as a biocatalyst is an interesting and appealing approach, especially being able to do so from crude cell lysate. I recommend this article for acceptance to Molecules following minor revision.

Originality/Novelty: 8/10

Significance: 8/10

Quality of Presentation: 7/10

Scientific Soundness: 7/10

Interest to the Readers: 9/10

Overall Merit: 8/10

English Level: 6/10

Minor Points:

Line 331, 344, 370, etc.: The authors use EDTA as the acronym for the dianhydride which is easily confused with the standard EDTA. The authors themselves are inconsistent on which name and acronym they use: EDTA, EDTA anhydride or EDTA dianhydride. All instances should be consistent throughout the article. Line 356: I believe the authors intended to indicate that weakly complexed “his-containing proteins” were eluted and not “his-tagged enzymes” In the version for review, Figures 1 and 2 have blurry / low resolution components. Figure 3: the legend again lists EDTA. Also the legend should be more clear that the second bar is EDTA dianhydride only and that the right-most bar is NPDGE with no EDTA dianhydride. Line 396: the test describes seven protein mixtures, yet only 6 entries are listed in Table 3 Figure 4: the compounds should be numbered and put the in same order that they appear in Figure 5. This will greatly enhance the effectiveness of Figure 5. Line 545: the total protein concentration applied to calculate the specific activity would include many non-active proteins. Additionally, the biocatalyst mass includes the mass of the MNPs and therefore, the activity would depend strongly on the functionalization completeness / mass of enzyme attached vs. mass of unfunctionalized MNP. Did the authors attempt to correct for such variations? The authors should supply the raw kinetic data (see below), as the mass percent of the PAL in the crude lystate was ~4 % from the methods, what was the mass % of the enzyme attached to the MNP?

Major Points:

The authors declare that they have no conflict of interests in the CoI statement. For full transparency, it is this reviewer’s opinion that the authors should declare their involvement in SynBioCat in this section. Reviewing SynBioCat, it is apparent that this research relates directly to the business activities of SynBioCat and further that the company would benefit from this report. Therefore, it appears as though both corresponding authors have vested financial interest in the publication of this research. I realize that the authors did list those positions in the affiliations, but ethically, it should be emphatically declared in the CoI section. The time course measurement or kinetics plots from the analysis should be provided as supplementary information. (See minor point 7 for more details).

Author Response

Answers of the questions of Reviewer 2

Manuscript ID: molecules-615889

Article title: “Fishing and hunting”– Selective immobilization of a recombinant phenylalanine ammonia-lyase from fermentation media

Minor Points:

1) Line 331, 344, 370, etc.: The authors use EDTA as the acronym for the dianhydride which is easily confused with the standard EDTA. The authors themselves are inconsistent on which name and acronym they use: EDTA, EDTA anhydride or EDTA dianhydride. All instances should be consistent throughout the article.

Response: We fully agree with the reviewer’s opinion and corrected the used names and abbreviations.

2) Line 356: I believe the authors intended to indicate that weakly complexed “his-containing proteins” were eluted and not “his-tagged enzymes” In the version for review, Figures 1 and 2 have blurry / low resolution components.

Response: Although we agree with the reviewer’s opinion, we think it is important to emphasize that the weakly complexed target proteins were eluted. We extended the sentence with the histidine-containing proteins. The figures were changed to more compression compatible types.

3) Figure 3: the legend again lists EDTA. Also the legend should be more clear that the second bar is EDTA dianhydride only and that the right-most bar is NPDGE with no EDTA dianhydride.

Response: In agreement with the reviewer’s opinion, we corrected the used abbreviations on the figure, and wrote a more detailed legend to the figure.

4) Line 396: the test describes seven protein mixtures, yet only 6 entries are listed in Table 3

Response: The mistyping has been corrected.

5) Figure 4: the compounds should be numbered and put the in same order that they appear in Figure 5. This will greatly enhance the effectiveness of Figure 5.

Response: We fully agree with the comment. The two figure were redesigned and mixed together using the same order of the multiepoxide compounds like in the diagram.

6) Line 545: the total protein concentration applied to calculate the specific activity would include many non-active proteins. Additionally, the biocatalyst mass includes the mass of the MNPs and therefore, the activity would depend strongly on the functionalization completeness / mass of enzyme attached vs. mass of unfunctionalized MNP. Did the authors attempt to correct for such variations? The authors should supply the raw kinetic data (see below), as the mass percent of the PAL in the crude lystate was ~4 % from the methods, what was the mass % of the enzyme attached to the MNP?

Response: An SDS-PAGE analysis was performed to prove the great selectivity of the bifunctional supports for the His-tagged PcPAL. Based on this test, it could be assumed that most of the immobilized proteins were the target enzyme. We always measured the activity of the cell lysate before and after the immobilization process and also of the imidazole contained elution fraction. Based on these activity measurements, the percent of the immobilized active PcPAL could be calculated. The target protein concentration in the cell lysate was determined from a purification process made with a commercially available Sepharose-Ni-NTA. From the known initial target protein concentration and activity (before immobilization), and the percentage residual PcPAL activity in the supernatant (directly after the immobilization) and in high concentrated imidazole (HIm) washing fraction (after elution washing) the immobilized yields were determined in each cases. All the additional information was included in the article or the new Supporting information.

Major Points:

7) The authors declare that they have no conflict of interests in the CoI statement. For full transparency, it is this reviewer’s opinion that the authors should declare their involvement in SynBioCat in this section. Reviewing SynBioCat, it is apparent that this research relates directly to the business activities of SynBioCat and further that the company would benefit from this report. Therefore, it appears as though both corresponding authors have vested financial interest in the publication of this research. I realize that the authors did list those positions in the affiliations, but ethically, it should be emphatically declared in the CoI section.

Response: The CoI part of the MS had been expanded with the clarifying of the interests of SynBiocat in this publication.

8) The time course measurement or kinetics plots from the analysis should be provided as supplementary information. (See minor point 7 for more details).

Response: A Supplementary information document had been edited to the MS and all above mentioned details had been added to it. The results explaining the calculation of immobilization and activity yields as Table S1 and the kinetic plots also had been added in Figure S4.

Round 2

Reviewer 1 Report

The authors have considered all the queries made by this reviewer and now, from my point of view, the manuscript is suitable for acceptance. I would only change one of the new sentences:

“One of them when the immobilization protocols were categorized into two main types, the physical and the chemical immobilizations, based on whether made chemical modification on the enzyme or not.”

into something like:

"In one of the, the immobilization protocols can be categorized into two main types, the physical and the chemical immobilizations, based on whether the enzyme was chemically modified or not".

Author Response

The main text has been corrected according to reviewer suggestion and marked with blue color in the uploaded re-revised manuscript.